# The Correlation between Spiritual Well-Being and Burnout of Teachers

## Hok-Ko Pong

Department of Business Management, Technological and Higher Education Institute of Hong Kong, Hong Kong, China; hkpong@thei.edu.hk

**Abstract:** This study examines the correlation between spiritual well-being and burnout symptoms, including emotional exhaustion (EE), depersonalisation (DP), and personal accomplishment (PA), among Chinese secondary school teachers in Hong Kong. The data were collected from 427 Chinese secondary school teachers (189 males, 238 females) aged 25–37 from different schools with one to eight years of teaching experience. The participants completed the Spiritual Health and Life-orientation Measure (SHALOM) to evaluate the status of their spiritual well-being in the personal and communal, environmental, and transcendental domains. The *Maslach Burnout Inventory*—Human Services Survey (*MBI-HSS*) was also used to measure the extent of burnout in the workplace. All domains of spiritual well-being were negatively associated with EE and DP, while the personal and communal domain and the transcendental domain of spiritual well-being were positively associated with PA. Multiple regression analysis revealed that all the specific domains of spiritual well-being explained 68.6% and 54.0% of the variance in teachers' EE and DP, respectively. Meanwhile, the same analysis found that the personal–communal and transcendental domains explained 74.9% of the variance in teachers' PA. The personal–communal domain of spiritual well-being was the strongest predictor of burnout.

**Keywords:** spiritual well-being; burnout; spirituality; teachers

## 1. Introduction

Teachers' spiritual health is directly and extensively correlated to students' learning, personal growth, and psychological well-being (Ismail et al. 2020). For instance, students can be influenced by the emotions and feelings, including joy, anger, sorrow, and sadness, of their teachers. Recently, most teachers undergo stress, exhaustion, and frustration, and such experience eventually leads to their decision to leave the profession. In relation to this experience, burnout has been determined as one of the main factors of the increase in employee turnover rate (Goodman and Boss 2002). Given this context, high teacher turnover rates incur losses to social and financial capital; moreover, the trend influences the morale of remaining colleagues and negatively affects the image of an organisation (Arokiasamy 2013). In addition, teachers' burnout and departure are detrimental to the development of students (Herman et al. 2020).

Maslach et al. (1996) asserted that burnout is a state of exhaustion, cynicism, and reduction in professional efficacy when a person is exposed to circumstances in which stress remains unreleased. It is also typically associated with one's physical, intellectual, social, emotional, and spiritual states (Brock and Grady 2002). As such, burnout has a huge impact on one's work performance.

Meanwhile, spirituality serves as a panacea that helps people to develop a tolerance for disasters and difficulties in life (Zohar et al. 2000). In one such example, Zhaleh and Ghonsooly (2017) examined primary school teachers and reported the role of high levels of spirituality in reducing job erosion. Previous studies have also determined a significant relationship between spiritual well-being and burnout among teachers (e.g., Zhaleh and Ghonsooly 2017; Ismail et al. 2020) and nurses and doctors (e.g., Sunaryo et al. 2017; Yang and Fry 2018).

Researchers have also conducted exhaustive studies on burnout (Chan and Hui 1995; Chan 2006) and spiritual well-being (Fisher and Wong 2013) in the context of Chinese teachers. Notably, many of these studies have found an association between these two aspects in the West. However, research on the correlation between the spiritual well-being and burnout of Chinese teachers in Asia remains scant.

The present research aims to enrich the related literature by investigating the link between spiritual well-being and burnout. Moreover, it explores how multiple dimensions can predict the role of spiritual well-being in job burnout. The current undertaking uses the findings of previous empirical studies to extend the examination of teachers in Asian regions within the Chinese cultural context. This investigation seeks to demonstrate the relationship between spiritual well-being and burnout using existing empirical research. Hence, this work addresses the following research questions:

Research Question 1: What is the relationships between spiritual well-being (including personal–communal, environmental, and transcendental domains) and burnout (including the dimensions of emotional exhaustion (EE), depersonalisation (DP), and personal achievement (PA)) of Chinese teachers?

Research Question 2: Is spiritual well-being considered a predictor of Chinese teachers' burnout?

### 1.1. Spirituality, Spiritual Health, and Spiritual Well-Being

As the essence of human experience, spirituality is one's search for and expression of the meaning of life (Zamaniyan et al. 2016). It refers to how an individual connects the self with others, nature, and the divine (Fisher 2021). Spirituality has always been tantamount to religious and divine-centred descriptions; however, this concept has also been expanded to describe one's beliefs and faith, search for meaning, and creation of experiences (McSherry et al. 2004). Moreover, a person always expresses spirituality through beliefs, values systems, traditions, and practices (Tanyi 2002).

Spirituality and religiosity have typically been defined interchangeably (Wilfred 2006; McSherry et al. 2004; Puchalski et al. 2014). However, spirituality and religiosity have certain distinctions. Notably, spirituality is a component of religion but not necessarily exclusively linked to it. As such, scholars have long debated about the correlation between the two concepts (Ammerman 2013), with no consensus thus far.

The World Health Organisation (WHO) marks spirituality as the fourth dimension of health (Sepúlveda et al. 2002). As such, spiritual health is an essential dimension of one's overall well-being (Eberst 1984). In emphasising the integration of the body, mind, and spirit, spiritual health calls for an environment of inner harmony to connect oneself with others, nature and a higher entity (Fisher 2021).

Spiritual well-being can indicate one's quality of life in terms of spirituality, and such a state of being has a pervading and integrating effect on other dimensions of life, including physical, psychological, and social health (Fisher 2010). Furthermore, it is linked to happiness and is often described as people's spiritual condition (Alvarez et al. 2016). Spiritual well-being also comprises the perceived state of spiritual inclusion and harmony (Pong et al. 2020).

Moreover, spiritual well-being refers to how one expresses his or her spirituality. This expression is measured in the personal and communal, environmental, and transcendental dimensions of well-being (Fisher 1998). Fisher (1998) initially developed the Spiritual Health and Life-orientation Measure (SHALOM). This measurement comprised 20 items that were equally distributed into the four dimensions of spiritual well-being.

In the personal domain, it refers to the meaning, purpose, and value of life. They are inner spirits such as self-esteem and uniqueness. In the communal domain, it refers to the wisdom of social relations, which includes love, fairness, respect, humility, and trust. Morals, values, and beliefs are included in the communal domain. In the environmental domain, it refers to the nurture and care of plants and animals, as well as the idea of living in harmony with the surrounding environment. In the transcendent domain, it refers to

the relationship with the Creator/Divine/Higher power. It focuses on the worship and adoration of religions and the mysteries of the universe.

The participants of the experiment had to provide two answers for each item. One response was their ideal answer, while the other relied on their experiences. These answers were used to examine the ideal and experiential values of each question. The item questions on people's lived experiences collectively formed the Spiritual Well-being Questionnaire (SWBQ). Furthermore, in his revised measure, Fisher (2013) included the deceased, a higher power, and the higher self as an attempt to expand SHALOM to various worldviews, ranging from belief in God to non-belief.

Scholars have extensively used Fisher's scales in various world contexts. This wide array of researchers includes Chinese (Pong et al. 2020), Hebrew (Elhai et al. 2018), Brazilian (Nunes et al. 2018), Portuguese (Valdivia et al. 2020), French (Papillon and Rajesh 2020), Lithuanian (Riklikiene et al. 2018), and Spanish (Muñoz-García and Aviles-Herrera 2014) scholars. Numerous works (e.g., Pong et al. 2020; Fisher 2021; Papillon and Rajesh 2020) have provided strong support for the validity and reliability of this psychometric test through exploratory factor analysis (EFA) and multi-group confirmatory factor analysis (CFA). In addition, researchers have determined the scale's high internal consistency, composite reliability and variance. Given this confirmation, the SWBQ has been used in the current study as well.

### 1.2. Burnout and Its Effects on Job Performance

Job burnout was initially proposed by Freudenberger (1974), who defined the concept as an individual's feelings of failure, exhaustion, and collapse resulting from the excessive demand for one's abilities, energy, and resources. It is a form of physical, psychological, and emotional exhaustion, ranging from chronic fatigue to negative perceptions and attitudes towards work, life, or others (Maslach et al. 2001). Thus, burnout is an indication as well as a symbol of negative emotions due to work stress; as such, burnout can occur based on how one's intrinsic characteristics, such as personality traits, react to environmental factors (Kokkinos 2007).

Whereas work stress is a temporary state and is experienced when an individual struggles to cope with pressing work demands, the imbalance resulting from burnout is long term (Schaufeli and Buunk 2003). Notably, job burnout does not necessarily occur when an individual has to fulfil numerous tasks within his or her capacity for an extended period. However, this form of distress develops when the requirements become too difficult to bear.

Maslach and Zimbardo's (1982) definition of burnout is the most widely accepted in the field. According to Maslach and Jackson (1981), job burnout has three levels, namely:

(1) Emotional exhaustion: At this stage, an individual experiences exhaustion, loss of interest, and powerlessness. The person struggles to cope physically and psychologically with work demands, particularly when interacting with others at work.
(2) Depersonalisation: The individual's interaction with others is characterised by negativity and indifference. At this stage, colleagues or customers are treated as objects.
(3) Lack of personal accomplishment: The individual's is dissatisfied with his or her work performance and loses overall motivation to work.

Burnout has multiple effects on workers, particularly in the physical, psychological, emotional, and behavioural domains. For instance, individuals experience physiological imbalances due to prolonged and heavy stress from work. Fatigue, headaches, colds, insomnia, weight loss, ulcers, urological illnesses, and cardiovascular illnesses are among the outcomes of such imbalance (Maslach 2001).

Meanwhile, the psychological effects of burnout can cause an individual to develop negative perceptions about others; moreover, mental abilities may become impaired, that is, they may start to lack empathy, self-confidence, self-worth, trust, and focus (Schaufeli and Greenglass 2001). Individuals also have the tendency to develop negative emotions when experiencing burnout. Among these negative feelings are anger, depression,

pessimism, anxiety, loss of inner peace, loneliness, nervousness, timidity, fear, helplessness, and despair. Moreover, individuals may manifest alienating behaviours, thereby affecting their interpersonal relationships and work attitudes (Schaufeli and Greenglass 2001). Some common symptoms are apathy, lack of pain, loss of patience, tendency to complain, unreasonable suspicions, work–family conflict, low productivity, lack of commitment, work alienation, absenteeism, and resignation (Schaufeli and Greenglass 2001).

### 1.3. Spirituality and Work Performance

Researchers such as Kim and Yeom (2018) and Akbari and Hossaini (2018) observed that spirituality has a significantly negative relationship with burnout among employees. The results of their studies suggested that higher spirituality decreases the likelihood of employee burnout and vice versa. Scholars have also determined how spirituality helps to prevent employees' burnout across various ethnic groups (Holland and Neimeyer 2005; Kaur et al. 2013; Ntantana et al. 2017). Moreover, research has shown that spirituality aids individuals in achieving self-development and career excellence (Altaf and Awan 2011). For instance, Van der Walt and De Klerk (2014) noted the positive impacts of spirituality on creativity, job satisfaction, team performance, and commitments within an organisation. Meanwhile, Kinjerski and Skrypnek (2006) found that higher spiritual health fosters employee loyalty and commitment. Therefore, spiritual health is the best indicator of employee performance.

Some studies (e.g., Schaufeli and Buunk 2003; Maslach et al. 1997; Kop and Euwema 2001) have observed that burnout diminishes performance and exacerbates turnover, absenteeism, and declined productivity. Under these circumstances, the negative effects and indicators of burnout manifest in an individual's spiritual state (Moradi et al. 2017).

### 1.4. The Measurement of Burnout—Maslach Burnout Inventory

Maslach and Jackson (1981) developed the *Maslach Burnout Inventory*—Human Services Survey (*MBI-HSS*) to measure burnout among employees. The 22 items in the questionnaire measure the three dimensions of emotional exhaustion (EE), depersonalisation (DP), and personal achievement (PA). The items are measured on a seven-point Likert scale, which ranges from 0 ('never') to 6 ('daily'). Questions on EE assess feelings of emotional overextension and exhaustion at work. Meanwhile, DP gauges the reactions, particularly unsympathetic and impersonal interactions, of individuals towards others in organisations. In addition, PA deals with one's proficiency, ability, and achievements at work. Hence, scores for the items for PA are inversed: items with higher scores entailed higher levels of burnout (i.e., EE and DP are higher, whereas PA is lower).

Researchers examining burnout have widely used the MBI in their studies (Bianchi et al. 2015). The latest version of the MBI, or the MBI-General Survey (MBI-GS), has improved the scales by relabelling the three dimensions of burnout, namely, exhaustion, cynicism, and professional efficacy (Maslach et al. 1996). In the current study, the *MBI-HSS* was selected given its credibility and wide usage in this field of research (Slabšinskienė et al. 2021).

The *MBI-HSS* has been fully validated and extensively used in a number of studies worldwide (Wickramasinghe et al. 2018; Schutte et al. 2000). Given such wide application, the questionnaire has been translated into multiple languages, including Chinese (Fong et al. 2014; Wang et al. 2020), German (Turhan et al. 2021), Dutch (Roelofs et al. 2005), Italian (Loera et al. 2014), and Korean (Oh and Lee 2009). Recent literature on the psychometric aspects of the questionnaire has indicated the suitability of the model's use of three domains compared with other available models (Turhan et al. 2021; Wang et al. 2020).

## 2. Method

### 2.1. Design, Research Instrument, Data Collection, and Procedure

Given the constraints of the pandemic, the study collected data through an online questionnaire. The questionnaire was divided into three sections. The first part collected demographic information, including gender and age. Then, the second section included

questions from the spiritual well-being questionnaire (SWBQ) developed by Fisher (1998, 2013). Lastly, the final section was the *MBI-HSS* developed by Maslach and Jackson (1981). Given that the cultural context of Hong Kong is bilingual, the questionnaire was available in Chinese and English, with participants being allowed to choose and complete either version. The questionnaire took about 15 min to answer. The cover page of the online version delineated the purpose of the research. Then, it explained that the identities of the respondents would remain anonymous and that the information they would share would be confidential. Moreover, the pre-questionnaire instructions also stated that the participants were free to terminate their involvement at any point of the questionnaire. The respondents also had to agree with the terms and provide a written informed consent to participate. This study was approved by the Research Ethics Committee of the affiliated institution. The online questionnaire started on 1 May 2021 and ended on 31 August 2021.

Chain-referral sampling was used, as this technique has been validated and used in numerous exploratory studies (Penrod et al. 2003). In this non-probability sampling technique, individuals who are initially screened to be part of the sample have to recruit additional respondents who fit the criteria of the survey (Heckathorn 2002). A group of part-time education students at the postgraduate level (Master's or postgraduate diploma) in a university in Hong Kong were invited to participate in an online questionnaire. Then, they were asked to forward the link for the online questionnaire to their colleagues and friends in the profession. Emails were sent through the help of lecturers and programme leaders to the targeted participants. The emails had a hyperlink that led to an online survey website. The survey included an informed consent form and the set of questionnaires. In addition, browser cookies were set up to prevent respondents from using the same browser to answer the questionnaire more than once.

## 2.2. Participants

A total of 446 questionnaires were received. However, 19 were excluded from the final analysis as the respondents failed to answer most of the questions. In the sample, 189 out of the 446 respondents represented 44.3% were male, with 238 being female. Most respondents were aged between 25 and 37 and had 1 to 8 years of teaching experience in schools.

## 2.3. Spiritual Well-Being Questionnaire (SWBQ)

The SWBQ section (Fisher 1998, 2013) has 20 statements, which asked participants to rate the importance of each item in everyday life (i.e., lived experience). Each item is scored using a five-point Likert scale, with 1 being 'strongly disagree' and 5 being 'strongly agree'. This approach helps to collect information on the four areas of spiritual well-being, namely, the personal (e.g., 'meaning in life'), communal (e.g., 'kindness towards other people'), environmental (e.g., 'oneness with nature'), and transcendental (e.g., 'prayer in life') domains. Each area has five items, thereby constituting the 20 statements in the psychometric test.

The original SWBQ has four domains: personal, communal, environmental, and transcendental (Fisher et al. 2000). However, the three-domain model is more suitable for research within the Chinese context; notably, the personal and communal domains have been merged on the basis of the outcomes of the factor analysis (Pong et al. 2020; Pong 2021). This integration is valid, as personal and communal domains have a strong correlation in Chinese culture (Hofstede 2001). This correlation was in line with the Chinese view on the inter-relatedness of personal cultivation and social harmony (Pong et al. 2020). It is because the Confucian tradition's emphases on personal cultivation to achieve social harmony (see The Great Learning; Legge 1971, pp. 357–58). In terms of the Cronbach's alphas of these three areas, the personal–communal domain is 0.989, the environmental domain is 0.972, and the transcendental domain is 0.966. In the current study, principal components analysis corroborated the aptness of the three-factor model. Additionally, the Kaiser–Meyer–Olkin value was 0.854, and Bartlett's test of sphericity was significant at $p < 0.001$. Meanwhile,

EFA identified three factors accounting for the personal–communal, environmental, and transcendental domains. Each area had an eigenvalue >1.0 and explained 46.621%, 22.059%, and 21.995% of the variance, respectively. Table 1 presents the factor loadings for the model.

**Table 1.** Results of exploratory factor analysis of the Spiritual Well-being Questionnaire items (SWBQ) (N = 427).

| | Rotated Component Matrix [a] | | |
|---|---|---|---|
| | **Component** | | |
| | **Personal and Communal** | **Environmental** | **Transcendental** |
| SWBQ1: A love of other people | **0.934** | −0.013 | 0.055 |
| SWBQ2: Personal relationship with the Divine/God | −0.003 | 0.010 | **0.979** |
| SWBQ3: Forgiveness toward others | **0.953** | −0.040 | −0.017 |
| SWBQ4: Connection with nature | −0.118 | **0.935** | −0.001 |
| SWBQ5: A sense of identity | **0.946** | −0.038 | 0.011 |
| SWBQ6: Worship of the Creator | −0.006 | 0.024 | **0.974** |
| SWBQ7: Awe at a breathtaking view | −0.049 | **0.961** | −0.004 |
| SWBQ8: Trust between individuals | **0.953** | −0.099 | 0.087 |
| SWBQ9: Self-awareness | **0.930** | −0.122 | 0.091 |
| SWBQ10: Oneness with nature | −0.096 | **0.963** | −0.017 |
| SWBQ11: Oneness with God | 0.071 | 0.073 | **0.941** |
| SWBQ12: Harmony with the environment | 0.000 | **0.975** | −0.007 |
| SWBQ13: Peace with God | 0.108 | −0.155 | **0.850** |
| SWBQ14: Joy in life | **0.924** | −0.083 | 0.054 |
| SWBQ15: Prayer in life | 0.037 | 0.007 | **0.962** |
| SWBQ16: Inner peace | **0.968** | −0.060 | 0.036 |
| SWBQ17: Respect for others | **0.976** | −0.035 | −0.007 |
| SWBQ18: Meaning in life | **0.966** | −0.025 | 0.006 |
| SWBQ19: Kindness toward other people | **0.963** | −0.063 | 0.042 |
| SWBQ20: A sense of 'magic' in the environment | −0.075 | **0.905** | −0.012 |

Note: Items loaded on each factor are in boldface. [a] 3 components extracted.

### 2.4. The Maslach Burnout Inventory—Human Services Survey (MBI-HSS)

It is a psychometric test with 22 items on the three dimensions, namely, EE, DP, and PA, which determine the extent of employee burnout. Nine items measure EE, with statements such as 'I feel burned out because of my work'. This domain has a sum score ranging 0–54. Five items measure DP, with statements such as 'I have a feeling that my colleagues blame me for some of their problems'. This section has a sum score of 0–30. Meanwhile, eight items measure PA, with statements such as 'I feel full of energy'. This area has a sum score ranging 0–48.

Moreover, the Cronbach's alphas for EE, DP, and PA are 0.987, 0.978, and 0.995, respectively. In the current study, principal components analysis demonstrated the suitability of the three-domain model in our sample. The Kaiser–Meyer–Olkin value was 0.945, and Bartlett's test of sphericity was significant at $p < 0.001$. In addition, the EFA identified three factors that corresponded to EE, DP, and PA. Each domain had an eigenvalue >1.0 and explained 75.431%, 10.646%, and 7.335% of the variance, respectively. Table 2 presents the factor loadings for the model.

**Table 2.** Results of exploratory factor analysis of the *Maslach Burnout Inventory*—Human Services Survey (*MBI-HSS*) items (N = 427).

| | Rotated Component Matrix [a] | | |
|---|---|---|---|
| | Component | | |
| | Emotional Exhaustion (EE) | Personal Accomplishment (PA) | Depersonalisation (DP) |
| 1. I feel emotionally exhausted because of my work | **0.869** | −0.377 | 0.275 |
| 2. I feel worn out at the end of a working day | **0.856** | −0.356 | 0.217 |
| 3. I feel tired as soon as I get up in the morning and see a new working day stretched out in front of me | **0.777** | −0.296 | 0.254 |
| 4. I can easily understand the actions of my colleagues/supervisors | −0.429 | **0.844** | −0.298 |
| 5. I get the feeling that I treat some clients/colleagues impersonally, as if they were objects | 0.264 | −0.278 | **0.871** |
| 6. Working with people the whole day is stressful for me | **0.838** | −0.411 | 0.255 |
| 7. I deal with other people's problems successfully | −0.418 | **0.851** | −0.302 |
| 8. I feel burned out because of my work | **0.843** | −0.406 | 0.258 |
| 9. I feel that I influence other people positively through my work | −0.407 | **0.851** | −0.306 |
| 10. I have become more callous to people since I have started doing this job | 0.208 | −0.252 | **0.931** |
| 11. I'm afraid that my work makes me emotionally harder | 0.207 | −0.218 | **0.925** |
| 12. I feel full of energy | −0.392 | 0.846 | −0.285 |
| 13. I feel frustrated by my work | **0.857** | −0.393 | 0.251 |
| 14. I get the feeling that I work too hard | **0.813** | −0.310 | 0.198 |
| 15. I'm not really interested in what is going on with many of my colleagues | 0.280 | −0.325 | **0.872** |
| 16. Being in direct contact with people at work is too stressful | **0.830** | −0.433 | 0.245 |
| 17. I find it easy to build a relaxed atmosphere in my working environment | −0.416 | **0.830** | −0.328 |
| 18. I feel stimulated when I been working closely with my colleagues | −0.398 | **0.853** | −0.309 |
| 19. I have achieved many rewarding objectives in my work | −0.481 | **0.777** | −0.278 |
| 20. I feel as if I'm at my wits' end | **0.863** | −0.368 | 0.290 |
| 21. In my work I am very relaxed when dealing with emotional problems | −0.407 | **0.852** | −0.300 |
| 22. I have the feeling that my colleagues blame me for some of their problems | 0.348 | −0.361 | **0.783** |

Note: Items loaded on each factor are in boldface. [a] 3 components extracted.

## 3. Results

This study employed SPSS version 25 to analyse the data collected from the questionnaire. Then, data cleaning was conducted to address coding errors and illogical values. In the end, the process had no missing data.

*Descriptive Statistics*

In the initial section of the questionnaire, participants had to provide demographic information, including their gender, age, and teaching experience. Table 3 presents the descriptive statistics. In the 427 valid responses of the respondents, 238 were female (55.7%) and 189 were male (44.3%). The respondents were predominantly aged 26–33, with 399 respondents or 93.5% belonging to this group. In addition, 285 of the participants or 66.7% had a teaching experience of about 2 to 4 years. The results of the examination of demographic differences on the overall scores of the SWBQ and MBI are in Table 3.

**Table 3.** Descriptive statistics: Participants' demographics and their relationship with spiritual well-being and burnout (*N* = 427).

| Factors | N (%) | Per and Com M (SD) | Environ M (SD) | Transcend M (SD) | EE M (SD) | DP M (SD) | PA M (SD) |
|---|---|---|---|---|---|---|---|
| All | 427 (100%) | 3.21 (0.63) | 2.22 (0.62) | 2.98 (0.39) | 2.21 (0.63) | 2.04 (0.59) | 1.55 (1.12) |
| **Gender** | | | | | | | |
| (1) Male | 189 (44.3%) | 3.19 (0.65) | 2.20 (0.63) | 3.00 (0.42) | 2.20 (0.63) | 2.09 (0.71) | 1.53 (1.15) |
| (2) Female | 238 (55.7%) | 3.23 (0.61) | 2.24 (0.62) | 2.97 (0.37) | 2.22 (0.64) | 2.00 (0.47) | 1.56 (1.11) |
| **Age** | | | | | | | |
| (1) 25 and below | 21 (4.9%) | 3.59 0.55 | 2.15 0.70 | 3.12 0.47 | 2.00 0.59 | 1.81 0.35 | 2.16 1.26 |
| (2) 26–29 years old | 172 (40.3%) | 3.17 0.67 | 2.25 0.61 | 2.97 0.42 | 2.23 0.67 | 2.07 0.67 | 1.52 1.19 |
| (3) 30–33 years old | 227 (53.2%) | 3.19 0.60 | 2.21 0.63 | 2.97 0.36 | 2.23 0.60 | 2.06 0.54 | 1.48 1.02 |
| (4) 34–37 years old | 7 (1.6%) | 3.57 0.39 | 2.20 0.76 | 3.14 0.38 | 1.70 0.70 | 1.63 0.44 | 2.57 1.27 |
| **Teaching experience** | | | | | | | |
| (1) Less than 2 years | 56 (13.1%) | 3.55 (0.56) | 2.38 (0.64) | 3.06 (0.47) | 1.87 (0.61) | 1.75 (0.41) | 2.25 (1.27) |
| (2) 2–4 years | 285 (66.7 %) | 3.16 (0.62) | 2.16 (0.62) | 2.98 (0.37) | 2.29 (0.59) | 2.10 (0.60) | 1.44 (1.05) |
| (3) 5–8 years | 86 (20.1%) | 3.14 (0.62) | 2.31 (0.61) | 2.95 (0.43) | 2.16 (0.69) | 2.04 (0.62) | 1.44 (1.10) |
| **Highest academic qualification achieved** | | | | | | | |
| (1) Bachelor's degree | 296 (69.3%) | 3.27 (0.52) | 2.17 (0.59) | 3.00 (0.35) | 2.17 (0.59) | 2.06 (0.60) | 1.62 (1.15) |
| (2) Postgraduate Diploma | 101 (23.7%) | 3.09 (0.58) | 2.34 (0.71) | 2.97 (0.45) | 2.30 (0.67) | 1.97 (0.44) | 1.36 (0.97) |
| (3) Master's degree | 30 (7.0%) | 3.01 (0.69) | 2.35 (0.59) | 2.79 (0.54) | 2.34 (0.84) | 2.15 (0.88) | 1.46 (1.25) |
| **Religious beliefs** | | | | | | | |
| (1) Yes | 259 (60.7%) | 2.96 (0.58) | 2.22 (0.66) | 2.93 (0.43) | 2.42 (0.60) | 2.17 (0.68) | 1.09 (0.88) |
| (2) No | 168 (39.3%) | 3.59 (0.50) | 2.21 (0.57) | 3.06 (0.31) | 1.88 (0.53) | 1.84 (0.33) | 2.25 (1.09) |

Note. *N* = 427. SWBQ = Spiritual Well-Being Questionnaire items. The SWBQ subscales are Per and Com = personal and communal; Environ = environmental; Transcend = transcendental. *MBI-HSS = Maslach Burnout Inventory*—Human Services Survey (*MBI-HSS*). The *MBI-HSS* subscales are EE = emotional exhaustion; DP = depersonalisation; PA = personal accomplishment.

Table 4 lists the scores on the two questionnaires along with their correlations. In terms of spiritual well-being, Chinese teachers' personal–communal domain, environmental domain, and transcendental domain were negatively and significantly associated with the EE and DP of burnout. Meanwhile, the personal–communal domain and transcendental domain had a positive correlation with PA.

**Table 4.** Pearson correlation between spiritual well-being and burnout.

| Career Adaptability | Spiritual Well-Being | | |
|---|---|---|---|
| | **Personal and Communal** | **Environmental** | **Transcendental** |
| EE | −0.693 ** | −0.263 ** | −0.332 ** |
| DP | −0.604 ** | −0.283 ** | −0.248 ** |
| PA | 0.840 ** | 0.067 | 0.282 ** |

Note: ** $p < 0.01$.

A significant and negative correlation was established between the personal–communal domain and EE and DP, particularly showing a moderate-to-strong negative correlation. Pearson's *r* values ranged from −0.604 to −0.693. Conversely, a positive and significant correlation was observed between personal–communal domain and PA, thus showing a strong positive correlation (*r* = 0.840).

Furthermore, a significant and negative correlation between the environmental and transcendental domains and EE and DP was observed, thus showing a weak negative correlation. Pearson's *r* values ranged from −0.248 to −0.332. Conversely, a positive and significant correlation was observed between the transcendental domain and PA (*r* = 0.282).

We used stepwise multiple regression analyses to assess the domains of spiritual well-being as predictor variables and the three factors of MBI as dependent variables. The three analyses are presented in Table 5.

**Table 5.** Results of hierarchical regression analyses with spiritual well-being (SWB) in the specific domains as predictors of participants' burnout.

| Variable | | *β* | T | F | R | R2 | Δ R2 | Adjusted R2 |
|---|---|---|---|---|---|---|---|---|
| **MBI–EE** | | | | | | | | |
| Model 1 | | | | 392.111 *** | 0.693 | 0.480 | 0.480 | 0.479 |
| | Personal and communal (SWB) | −0.693 | −19.802 | | | | | |
| Model 2 | | | | 329.275 *** | 0.780 | 0.608 | 0.128 | 0.606 |
| | Personal and communal (SWB) | −0.741 | −24.156 | | | | | |
| | Environment (SWB) | −0.362 | −11.792 | | | | | |
| Model 3 | | | | 307.552 *** | 0.828 | 0.686 | 0.077 | 0.683 |
| | Personal and communal (SWB) | −0.717 | −25.992 | | | | | |
| | Environment (SWB) | −0.365 | −13.274 | | | | | |
| | Transcendental (SWB) | −0.279 | −10.200 | | | | | |
| **MBI–DP** | | | | | | | | |
| Model 1 | | | | 244.031 *** | 0.604 | 0.365 | 0.365 | 0.363 |
| | Personal and communal (SWB) | −0.604 | −15.621 | | | | | |
| Model 2 | | | | 211.355 *** | 0.707 | 0.499 | 0.134 | 0.497 |
| | Personal and communal (SWB | −0.653 | −18.835 | | | | | |
| | Environment (SWB) | −0.370 | −10.671 | | | | | |
| Model 3 | | | | 165.413 *** | 0.735 | 0.540 | 0.041 | 0.537 |
| | Personal and communal (SWB | −0.636 | −19.050 | | | | | |
| | Environment (SWB) | −0.373 | −11.194 | | | | | |
| | Transcendental (SWB) | −0.202 | −6.109 | | | | | |
| **MBI–PA** | | *β* | T | F | R | R2 | Δ R2 | Adjusted R2 |
| Model 1 | | | | 1014.778 *** | 0.840 | 0.705 | 0.705 | 0.704 |
| | Personal and communal (SWB) | 0.840 | 31.856 | | | | | |
| Model 2 | | | | 633.647 *** | 0.866 | 0.749 | 0.044 | 0.748 |
| | Personal and communal (SWB | 0.821 | 33.660 | | | | | |
| | Transcendental (SWB) | 0.212 | 8.674 | | | | | |

*** $p < 0.001$.

The personal–communal domain was used as the predictor for EE in Model 1. The results were as follows: $F(1, 425) = 392.111$, $p < 0.001$, thus accounting for 48.0% of variance in EE. The environmental domain was added in Model 2, and the results are $F(2, 424) = 329.275$, $p < 0.001$. Thus, they accounted for 60.8% of the variance and explained an additional 12.8% in EE. Then, the transcendental domain added used in Model 3, with

the following outcomes: $F$ (2, 423) = 307.552, $p < 0.001$. They accounted for 68.6% of the variance and explained an additional 7.70% in EE.

For DP, the corresponding values for Model 1 were $F$(1, 425) = 244.031, $p < 0.001$. They accounted for 36.5 % of the variance. Then, the environmental domain was added in the equation for Model 2: $F$(2, 424) = 211.355, $p < 0.001$, thereby accounting for 49.9% of the variance and explaining an additional 13.4% in DP. The transcendental domain was added in Model 3, with outcomes of $F$(2, 423) = 165.413, $p < 0.001$. They accounted for 54.0% of the variance and explained an additional 4.1% in DP.

For PA, the personal–communal domain was used as the predictor in Model 1, with outcomes of $F$(1, 425) = 1014.778, $p < 0.001$. They accounted for 70.5% of the variance in PA. The environmental domain was entered into the equation in Model 2, with the following results: $F$(2, 424) = 633.647, $p < 0.001$. They accounted for 74.9% of variance and explained an additional 4.4 % in PA. In three sets of analyses, the personal–communal domain was the strongest predictor of burnout.

## 4. Discussion

The findings of the current study are consistent with those of Kim and Yeom (2018) and Akbari and Hossaini (2018), who asserted that higher levels of spiritual well-being and spiritual health, respectively, have a significant correlation in the decrease of employee burnout. Kim and Yeom (2018) further stated that employees' internal peace and psychological stability may buffer the influence of job-related stress on burnout. In addition, another study found that developing higher levels of spiritual well-being is a preventive strategy to reduce burnout (Estupiñan and Kibble 2018).

Estupiñan and Kibble (2018) demonstrated that strong spiritual resources and daily activities, such as prayers, increase one's life satisfaction and diminish psychological distress and burnout. Akbari and Hossaini (2018) also found that emotion regulation adequately mediates between spirituality and burnout. Moreover, in relation to the current study, Ismail et al. (2020) also established that spirituality predicts burnout among teachers.

### 4.1. Personal–Communal Domain

The findings are also consistent with those of Zhaleh and Ghonsooly (2017). They observed a significant correlation between the meaning and purpose of life and burnout. The researchers also found that higher levels of awareness, self-identity, and self-confidence lower people's likelihood of leaving their profession. These individuals may even see their career pursuits as a calling. Furthermore, King (2008) revealed that individuals who have the capacity to create meaning and purpose in their lives develop the ability to identify the causes of workplace stress and conflict, thereby being able to find solutions rather than exacerbate the problems.

Oglesby et al. (2021) also found that elements that strengthen interpersonal relationships, such as forgiveness, love, and concern for others, significantly help employees raise awareness of their negative emotions and subsequently aid in managing these emotions to heal the symptoms of burnout. The researchers also stated that spirituality serves as a protective factor to alleviate conflicts between work and family life, emotional exhaustion, depersonalisation, diminished personal accomplishment, substance abuse, and resignation from work. Their study also demonstrated the high correlation between greater perceived social support and harmony with others with lower levels of burnout.

The findings of the current study are also consistent with those of Hardiman and Simmonds (2013). These researchers observed that having a clear purpose in life and maintaining good interpersonal relationships are significant indicators of low emotional exhaustion and depersonalisation and high personal accomplishment. In addition, Kim and Yeom (2018) and Tasharrofi et al. (2013) determined that those with high spiritual well-being increase their tolerance for difficult situations. Therefore, symptoms of burnout are less likely to manifest in these individuals.

*4.2. Environmental Domain*

The findings of this study are consistent with those of van den Berg and Beute (2021), who determined that connecting with nature and fostering environmental awareness have a negative correlation with burnout. In the past decade, researchers (e.g., Craig and Prescott 2017; Pietilä et al. 2015) have observed the positive effects of green environments and plant diversity on people's psychological and spiritual health. These scholars have demonstrated that plants and nature have calming and placebo effects on people. The research conducted by James et al. (2015) specifically showed that nature (green spaces and plants) plays a significant role in the recovery, resilience, and alleviation of psychological distress, depression, and anxiety.

Nature also has a substantial impact on creativity and emotional recovery. This impact can be observed in the functional changes in physical and mental fatigue (Yu et al. 2020). Moreover, the bond that forms between nature and people helps elevate feelings of satisfaction and value (Williams and Vaske 2003).

The findings of the study are also consistent with those of van den Berg and Beute (2021), who indicated the relationship of high environmental awareness and strong connection to nature with burnout. Specifically, those who enjoy nature develop a strong bond with it, which in turn decreases the likelihood of experiencing burnout symptoms, including emotional instability, fatigue, and indifference.

*4.3. Transcendental Domain and Burnout*

The findings of the study correspond to those of Kovács and Kézdy (2008), who determined that a statistically significant negative correlation exists between religious spirituality and burnout. Chirico (2017) explained that attending religious meetings regularly, receiving spiritual guidance and going to confession foster self-reflection. Moreover, they aid people in acknowledging their sins, which subsequently provides them a sense of release and may help individuals strike a balance between mind and body (Kutcher et al. 2010). Prayer also supports perceived control, which alleviates anxious thoughts and enhances people's adaptability to unpredictable events (Koenig 2013). Prayer also helps integrate the psychological and emotional levels of an individual (Chirico 2017).

The significance of one's suffering also pervades religious teachings. These teachings help people acknowledge suffering as a part of the human experience, thus adding value and meaning to suffering (Koenig 2013). As Koenig (2013) asserted, people gain a better understanding of the value of their work through their religious beliefs, thus creating space to integrate work into other aspects of life. Furthermore, Chirico (2017) found that religiosity protects people from burnout. Religion emphasises the importance of self-transcendence, which can be attained through working towards a cause or loving another (Kovács and Kézdy 2008). Through such beliefs, status, success, and income become less significant in defining individuals (Chirico 2017). Subsequently, these perspectives and the pursuit of a higher purpose alleviate stress.

The findings of this study were consistent with those of Kovács and Kézdy (2008), who concluded that religious practices, can predict employee burnout. For example, devout people are less likely to perceive that they are experiencing burnout symptoms, such as frequent mood swings, fatigue, and lack of empathy (Koenig 2013).

## 5. Limitations

The study has three major limitations. Firstly, the current study was conducted on in-service teachers, most of whom were between the ages of 26 and 33 (93.5%) and also the majority (60.7%) were religious. Therefore, the generalisation of the current findings may be biased. Future research should include a balanced number of age and religion with greater diversity in the sample to be more generalisable and avoid selection bias.

Secondly, the data were collected from participants using one-time point self-reported questionnaires. Due to the study design, long-term relationships among variables cannot be formed. We also cannot conclude the cause-and-effect relationship between the variables

without an experimental study. Therefore, hypothesis of causation should be considered carefully. Follow-up longitudinal studies will help address this limitation, so future studies using a prospective design are needed to confirm our findings.

Thirdly, there is a lack of other relevant psychometric tests, including emotional intelligence, adversity quotients, and personality traits, which are closely related to the psychosocial variables of in-service teachers. It is suggested that these be incorporated into future research.

## 6. Conclusions

This study reveals that Chinese teachers with high spiritual well-being likely experience less burnout. In addition, the findings demonstrate that the specific domains of spiritual well-being have a negative correlation with EE and DP. Meanwhile, the personal–communal domain and the transcendental domain demonstrate a positive correlation with DP. Notably, similar studies have been conducted within the Western context, such as those of Ismail et al. (2020) and Chirico et al. (2020). However, the current study is the first undertaking to examine the relationship between spiritual well-being and burnout among Chinese teachers in Asian regions. The findings of this research also have important implications for incorporating spiritual education into the curriculum of teacher training and human resource policies of schools. Doing so can protects employees from burnout and hopefully increases their feelings of belongingness, loyalty, and inclusion.

**Funding:** This research received no external funding.

**Informed Consent Statement:** Informed consent was obtained from all subjects involved in the study.

**Data Availability Statement:** Not applicable.

**Acknowledgments:** I am grateful for the valuable comments and useful advice from the two anonymous reviewers and editors. I would like to thank all participants in this study.

**Conflicts of Interest:** The author declare no conflict of interest.

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
