# Peer review of "The Correlation between Spiritual Well-Being and Burnout of Teachers"

_religions, doi:10.3390/rel13080760_

Round 1
Reviewer 1 Report
This study was well designed and executed. It was founded on a sound theoretical base, employed appropriate statistical methods of analysis to interrogate the hypotheses, which led to an interesting and informative report.
A few minor points need to be addressed to polish this presentation to make it ready for publication:
Line 68 McSherry (capital S)
Line 118 ‘work stress is a temporary state…’
Line 200 allowed to choose (rather than accomplish) either version
Line 274 This study employed (rather than ‘adopted’) SPSS
Lines 321, 323, 327 & 329 …domain was added (instead of just ‘used’)
Line 337 in three sets of analyses
Line 384 include detail on Pietila et al in the reference list
References need to be written in consistent type style and size, with capitals used appropriately for journal titles as well as names of instruments. Journal titles should be in italics.
Add https://... addresses to references.
Author Response
Dear Reviewer 1,
Thank you very much for your valuable comments and suggestions. I have revised the manuscript based on these helpful suggestions. I learned a lot from the revision. The list for revision is attached.
Best Regards,
Author

Reviewer 2 Report
This is an interesting paper and presents research that provides a contribution to the field. It does need some changes made.
1.The research questions need reworking. They lack clarity. The second one in particular reads in a rather circular manner.
2. The limitations need re-writing. The first and third limitations can be said about any study using a questionnaire methodology so are not worth stating. The second point could be a discussion point earlier on on the paper but it is not really a limitation but a critique of concepts.
3. The langague slips around a lot. Ensure that you use the term 'questionnaire' not 'survey' and when you refer to MBI and SHALOM they are psychometric tests with 3 and four scales respectively.
4. More is needed about Fishers theory before you move into the operationalisation of that theory.
5. I understand why you have pulled two aspects of Fisher's theory into one and it makes sense but you need to make the justification a little more strongly.

Author Response
Dear Reviewer,
Thank you very much for your valuable comments and suggestions. I have revised the manuscript based on these helpful suggestions. I learned a lot from the revision. The list for revision is given below. The PDF file with responses to the comments is attached.
Best Regards,
Author
|
Comments from Reviewer 2 |
|
Responses from author to reviewer 2 |
|
1. The research questions need reworking. They lack clarity. The second one in particular reads in a rather circular manner. |
|
Agreed and revised. |
|
2. The limitations need re-writing. The first and third limitations can be said about any study using a questionnaire methodology so are not worth stating. The second point could be a discussion point earlier on on the paper but it is not really a limitation but a critique of concepts. |
|
Agreed and revised. |
|
3. The langague slips around a lot. Ensure that you use the term 'questionnaire' not 'survey' and when you refer to MBI and SHALOM they are psychometric tests with 3 and four scales respectively. |
|
Agreed and revised. |
|
4. More is needed about Fishers theory before you move into the operationalisation of that theory. |
|
Agreed and revised in literature review. |
|
5. I understand why you have pulled two aspects of Fisher's theory into one and it makes sense but you need to make the justification a little more strongly. |
|
Agreed and revised. |
|
6. Comments and suggestions in PDF file |
|
Revised. |
